# Optimal protamine dosing after cardiopulmonary bypass: The PRODOSE adaptive randomised controlled trial

**Lachlan F. Miles**[1,2]\*, **Christiana Burt**[3], **Joseph Arrowsmith**[3], **Mikel A. McKie**[4], **Sofia S. Villar**[4], **Pooveshnie Govender**[3], **Ruth Shaylor**[2], **Zihui Tan**[3], **Ravi De Silva**[5], **Florian Falter**[3]

**1** Department of Critical Care, The University of Melbourne, Melbourne, Australia, **2** Department of Anaesthesia, Austin Health, Melbourne, Australia, **3** Department of Anaesthesia and Intensive Care, Royal Papworth Hospital NHS Foundation Trust, Cambridge, United Kingdom, **4** MRC Biostatistics Unit, School of Clinical Medicine, University of Cambridge, Cambridge, United Kingdom, **5** Department of Surgery, Royal Papworth Hospital NHS Foundation Trust, Cambridge, United Kingdom

\* lachlan.miles@unimelb.edu.au

## Abstract

### Background

The dose of protamine required following cardiopulmonary bypass (CPB) is often determined by the dose of heparin required pre-CPB, expressed as a fixed ratio. Dosing based on mathematical models of heparin clearance is postulated to improve protamine dosing precision and coagulation. We hypothesised that protamine dosing based on a 2-compartment model would improve thromboelastography (TEG) parameters and reduce the dose of protamine administered, relative to a fixed ratio.

### Methods and findings

We undertook a 2-stage, adaptive randomised controlled trial, allocating 228 participants to receive protamine dosed according to a mathematical model of heparin clearance or a fixed ratio of 1 mg of protamine for every 100 IU of heparin required to establish anticoagulation pre-CPB. A planned, blinded interim analysis was undertaken after the recruitment of 50% of the study cohort. Following this, the randomisation ratio was adapted from 1:1 to 1:1.33 to increase recruitment to the superior arm while maintaining study power. At the conclusion of trial recruitment, we had randomised 121 patients to the intervention arm and 107 patients to the control arm. The primary endpoint was kaolin TEG r-time measured 3 minutes after protamine administration at the end of CPB. Secondary endpoints included ratio of kaolin TEG r-time pre-CPB to the same metric following protamine administration, requirement for allogeneic red cell transfusion, intercostal catheter drainage at 4 hours postoperatively, and the requirement for reoperation due to bleeding. The trial was listed on a clinical trial registry (ClinicalTrials.gov Identifier: NCT03532594).

Participants were recruited between April 2018 and August 2019. Those in the intervention/model group had a shorter mean kaolin r-time (6.58 [SD 2.50] vs. 8.08 [SD 3.98] minutes; $p$ = 0.0016) post-CPB. The post-protamine thromboelastogram of the model group

**Data Availability Statement:** The data that support the findings of this study are available on request from the Royal Papworth Hospital NHS Foundation Trust Department of Research and Development

(papworth.randdadmin@nhs.net). The data cannot be made publicly available due to their containing information that could compromise the privacy of research participants.

**Funding:** Funding was awarded to LFM, CB, JA, RS and FF to facilitate the trial by the Jon Moulton Charity Trust (http://www.perscitusllp.com/moulton-charity-trust/). SSV is supported by a grant from the UK Medical Research Foundation (grant number MC_UU_00002/15). The funders had no role in study design, data collection and analysis, decision to publish, or preparation of the manuscript.

**Competing interests:** I have read the journal's policy and the authors of this manuscript have the following competing interests: FF has received speaking fees and research support from Abbott Laboratories and LivaNova. He is a member of the Scientific Advisory Committees for Abbott Point-of-Care, Werfen and LivaNova. The other authors have declared that no competing interests exist.

**Abbreviations:** ACT, activated clotting time; aPTT, activated partial thromboplastin time; CONSORT, Consolidated Standards of Reporting Trials; CPB, cardiopulmonary bypass; CT, clotting time; ICU, intensive care unit; INTEM, intrinsic rotational thromboelastometry; MA, maximum amplitude; PT, prothrombin time; TEG, thromboelastography.

was closer to pre-CPB parameters (median pre-CPB to post-protamine kaolin r-time ratio 0.96 [IQR 0.78–1.14] vs. 0.75 [IQR 0.57–0.99]; $p < 0.001$). We found no evidence of a difference in median mediastinal/pleural drainage at 4 hours postoperatively (140 [IQR 75–245] vs. 135 [IQR 94–222] mL; $p = 0.85$) or requirement (as a binary outcome) for packed red blood cell transfusion at 24 hours postoperatively (19 [15.8%] vs. 14 [13.1%] $p = 0.69$). Those in the model group had a lower median protamine dose (180 [IQR 160–210] vs. 280 [IQR 250–300] mg; $p < 0.001$).

Important limitations of this study include an unblinded design and lack of generalisability to certain populations deliberately excluded from the study (specifically children, patients with a total body weight >120 kg, and patients requiring therapeutic hypothermia to <28˚C).

## Conclusions

Using a mathematical model to guide protamine dosing in patients following CPB improved TEG r-time and reduced the dose administered relative to a fixed ratio. No differences were detected in postoperative mediastinal/pleural drainage or red blood cell transfusion requirement in our cohort of low-risk patients.

## Trial registration

ClinicalTrials.gov Unique identifier NCT03532594.

## Author summary

### Why was this study done?

- Mathematical models of heparin clearance have been postulated to reduce protamine dosing after cardiopulmonary bypass (CPB) and may improve clotting after cardiac surgery.

- Best practice guidelines have been unable to provide firm recommendations for the use of these models due to a lack of randomised clinical trial data.

- The PRODOSE trial was designed to address this gap in the evidence.

### What did the researchers do and find?

- We tested a mathematical model designed to calculate a bespoke protamine dose in patients undergoing cardiac surgery with CPB who were at a relatively low risk of bleeding, comparing it to a fixed dose ratio that is currently used widely (1 mg of protamine for every 100 IU of heparin given to establish safe anticoagulation before CPB, commonly known as a 1:1 fixed dose ratio).

- We found that using this mathematical model improved point-of-care measures of coagulation and reduced the amount of protamine given by 36.6% relative to the 1:1 fixed dose ratio.

- We did not find a difference in postoperative bleeding or use of blood products between the groups. However, as this is a Phase II trial, the study was not designed to accurately assess clinical outcomes, with an emphasis instead on safety and biochemical efficacy.

### What do these findings mean?

- Our findings suggest that mathematical models of heparin clearance can be safely used to dose protamine in the population studied and that dosing protamine according to the commonly used 1:1 fixed dose ratio is probably excessive in this population.

- Clinicians could consider using PRODOSE or similar models in patients at lower risk of bleeding after cardiac surgery in place of a 1:1 fixed dose ratio.

- Further research with a focus on clinical outcomes and a population at a higher risk of bleeding is warranted.

## Introduction

Adequate anticoagulation for cardiopulmonary bypass (CPB) is generally achieved using high doses of unfractionated heparin and is reversed using protamine sulphate [1]. Protamine can exert paradoxical anticoagulant (particularly inhibiting factor V activation and thrombin) and antiplatelet effects when administered in excess to circulating heparin [2]. Higher doses of protamine have been shown to increase the risk of transfusion and postoperative bleeding following cardiac surgery [3,4]. It is therefore prudent to dose protamine in such a way as to minimise these effects, while also ensuring effective heparin reversal.

Approaches to protamine dosing vary widely [5,6]. A common recommendation is a fixed 1:1 (1-mg protamine to every 100 IU of heparin) ratio based on the initial dose of heparin required to establish therapeutic anticoagulation [1,7]. This method does not directly account for heparin clearance and may lead to excessive protamine dosage [8]. Mathematical models have been proposed to enable the clinician to estimate the amount of circulating heparin at any time point based on common preoperative covariates, and thereby guide protamine administration. While small, nonrandomised studies of different models have shown that these approaches may improve coagulation and reduce protamine dose, uptake has been limited by a lack of prospective, randomised data [9]. To date, only 1 appropriately powered randomised trial has been published [10], and best practice guidelines continue to recommend the use of fixed dose ratios [11].

Following a pilot study [12], and having studied similar work [4,13], we hypothesised that protamine dosing guided by a 2-compartment pharmacokinetic derivation of heparin clearance would, relative to a conventional, fixed 1:1 ratio: (a) result in superior viscoelastic metrics of coagulation in the post-CPB period in patients undergoing cardiac surgery; and (b) use less protamine to achieve satisfactory haemostasis.

## Methods

### Trial design

We performed a 2-stage, adaptive randomised superiority trial. Patients in the first stage of the trial were randomised with equal probability using a blocked randomisation procedure to the

intervention (PRODOSE model) or control (fixed 1:1 ratio) arms. An interim analysis was scheduled to check for safety, consider a predefined futility rule, and an adaptation of the randomisation ratio based on the primary outcome data at that point aimed at maximising the number of patients in the superior arm while maintaining study power. The trial was performed at 2 centres—a dedicated cardiothoracic hospital in the United Kingdom and a tertiary, university-affiliated hospital in Australia. Ethical approval was received from the UK Health Research Authority (17/EE/0460) and the Austin Health Human Research Ethics Committee (HREC/17/Austin/566). The trial was registered (ClinicalTrials.gov Identifier: NCT03532594). All participants gave written informed consent prior to any study-related procedures.

## Study population

All patients presenting for cardiac surgery requiring CPB were considered for inclusion. Warfarin or novel oral anticoagulant therapy was ceased 5 or 3 days, respectively, prior to operation. Patients were excluded if they were <18 years old, had a total body weight >120 kg (due to unpredictable heparin requirements in obese individuals) [14], were dialysis dependent, had a known blood dyscrasia or platelet dysfunction, had received adenosine diphosphate receptor antagonists within 7 days of surgery, had received an unfractionated heparin infusion or therapeutic low molecular weight heparin <24 hours before surgery, required emergency surgery (defined as operation before the beginning of the next working day after the decision to operate was made), had an operative plan requiring therapeutic hypothermia to <28˚C, had complex surgical requirements (redo sternotomy, surgery on aortic arch, or descending thoracic aorta), or were undergoing solid organ transplantation.

## Coagulation management and conduct of cardiopulmonary bypass

A 2-g dose of tranexamic acid was administered to all patients prior to or with heparin. The initial dose of heparin (heparin sodium, Fannin (UK), Measham, UK or Pfizer Australia, Sydney, Australia) was 300 IU/kg based on total body weight, with a target activated clotting time (ACT) of ≥400 seconds. Another 5,000 IU of heparin was added to the pump prime. CPB was undertaken using flow rates of 2.2 to 2.5 min/m$^2$. Nasopharyngeal temperature was managed according to surgical preference (within the limitations of the protocol). The ACT was repeated every 30 minutes, and additional heparin was administered if this was ≤400 seconds. The administration of blood products in the operating room and for the first 24 hours postoperatively was based on standard institutional practices, codified to promote consistency (S1 Appendix).

## Randomisation and intervention

Participants were randomised immediately prior to or following induction of anaesthesia. The anaesthetist and surgeon were not blinded. Randomisation was implemented using a web-based portal (Sealed Envelope, London, UK) and a random permuted block algorithm with a block of size 4, stratified by centre to target the randomisation ratio established for each of the trial stages. In the first stage of the trial, participants were randomised in a 1:1 ratio. At the interim analysis, the randomisation ratio was adapted using primary outcome data up to that point, following our design based on methods first described by Zhang and Rosenberger and extended by simulations to accommodate violation of design assumptions (S2 Appendix) [15]. These predetermined rules aimed to maximise the number of patients in the treatment arm trending towards statistical superiority, ensuring that most patients received better treatment while preserving study power.

The control group received protamine post-CPB at a fixed dose ratio of 1 mg for every 100 IU of unfractionated heparin administered in the pre-CPB period to obtain the target ACT. This ratio was chosen as it is supported by expert guidance [11]. The intervention group received protamine dosed according to the amount of residual heparin calculated to remain at the end of CPB according to the PRODOSE model. Further doses of protamine were permitted in the operating theatre or within 4 hours of arrival in the intensive care unit (ICU) following the initial dose if thromboelastography (TEG) suggested residual heparinisation was present or to correct for residual heparin in unprocessed pump blood administered to the participant. Routine cell salvage was not employed at either institution for the management of the recruited patient population.

## The PRODOSE model

We previously performed a pilot study to validate a single-compartment pharmacokinetic model that predicted elimination of heparin corrected for ideal body weight [12]. This was then combined with the second compartment of a model described by Meesters and colleagues [13] to yield the following equation:

$$C_t = (C_0 \times A \times e^{-\alpha t}) + \left(C_0 \times B \times e^{-\left[\frac{\ln 0.5}{26+0.323\left(\frac{Heparin\ in\ system}{Ideal\ Body\ Weight}\right)}\right]t}\right),$$

where A = 0.1, α = 10, and B = 0.9. The equation was imported into Microsoft Excel (Microsoft, Redmond, Washington, United States of America). Ideal body weight was calculated using the Devine formula [16]. If total body weight was less than the ideal body weight, the former was used. The quantity of heparin in circulation at a given point in time ($C_t$) is calculated based on the initial loading dose or the amount of heparin calculated to be present at the time after the immediately previous administration ($C_0$) and the time elapsed since the last administration (t). At the conclusion of CPB, the final quantity of heparin within the system was calculated, and a protamine dose was administered at a ratio of 1-mg protamine for every 100 IU of heparin remaining.

## Laboratory and point-of-care investigations

Baseline investigations were performed following induction of anaesthesia, but prior to systemic heparinisation. These were the following: kaolin ACT, kaolin and heparinase TEG, full blood examination, prothrombin time (PT), and activated partial thromboplastin time (aPTT). TEG parameters were measured using the TEG 5000 or TEG 6s devices (Haemonetics, Braintree, Massachusetts, USA). These investigations were repeated 3 minutes after protamine administration following separation from CPB to ensure complete circulation of protamine before sampling was performed.

## Primary and secondary outcomes

The primary outcome was kaolin TEG r-time at 3 minutes post-protamine administration. This endpoint was chosen as it is far more sensitive to protamine-induced in vitro coagulation abnormalities than ACT [17], is a commonly used point-of-care test in clinical practice, and was the primary endpoint in the previous study we most closely sought to replicate [13]. Secondary outcomes were ratio of kaolin TEG r-time to heparinase TEG r-time at 3 minutes post-protamine administration, ratio of kaolin TEG r-time pre-CPB to kaolin TEG r-time at 3 minutes post-protamine administration, ratio of kaolin ACT pre-CPB to kaolin ACT at 3 minutes

post-protamine administration mediastinal/pleural drainage 4 hours postoperatively, and requirement for allogeneic red blood cell transfusion in the first 24 hours postoperatively. Return to theatre for bleeding in the first 24 hours following the procedure was included as a safety endpoint. Time points for postoperative measurement of mediastinal/pleural drainage, transfusion requirement, and return to theatre were based on previous studies [18]. Observed outcomes beyond these times were unlikely to be primarily related to heparin/protamine interactions at the time of surgery.

## Sample size calculation

The sample size was derived using summary measures of intrinsic rotational thromboelastometry (INTEM) clotting time (CT) as a TEG r-time surrogate from a comparable population [13]. Using a control post-protamine INTEM CT of 4.2 minutes, a standard deviation of 1.27 minutes, and a predicted effect size of 15%, simulations of the proposed design showed that a total sample size of 212 participants would result in type I and II error rates of 3.6% and 10%, respectively. The equivalent mean TEG r-time of the control group when transformed from INTEM CT was predicted to be 8 minutes with a standard deviation of 2.5 minutes [19]. To allow for missing data caused by changes of surgical plan leading to exclusion of a randomised participant, lack of follow-up data, or equipment malfunction, we recruited 228 participants, with an initial randomisation ratio of 1:1 between intervention and control groups.

## Interim analysis

Interim analysis was planned after recruitment of 50% ($n = 114$) of the study cohort. At this point, recruitment was paused, and data were considered. Specifically, the trial would be stopped if the return to theatre rate was significantly increased in the intervention arms or the t-statistic observed in the primary endpoint was higher than the prespecified futility boundary ($u_1 = 1.8$). If the primary outcome did not exceed the limits for safety or futility, then the preplanned adaptation of the randomisation ratio was to be done according to closed-form formulae in case of normality, while in the case of deviations from normality, the randomisation ratio would be adapted to 1:1.33, in favour of the beneficial treatment arm only if the observed treatment difference at the interim was half the predicted significant difference (7.5%). If the difference was less than 7.5%, the randomisation ratio would be maintained at 1:1 under deviations from normality. This is explained further in the Supporting information appended to this manuscript (S2 Appendix). Deviations from normality were determined through a significant ($p < 0.05$) Shapiro–Wilk test, and an observed difference exceeding the 7.5% threshold was observed. This triggered the randomisation rate that ensured more patients received the treatment arm, while the study power of 90% was preserved under assumptions of an exponential distribution. To minimise bias, only the primary endpoint and safety variables were analysed. The interim and final analyses were performed by 2 different statisticians. The clinical investigators remained blinded to any change of randomisation strategy.

## Statistical analysis

Primary efficacy analysis was carried out through a re-randomisation–based method, both ensuring type I error was preserved despite deviations in assumptions (that could not be corrected by transformation) and mitigating potential bias introduced by the adaptive design in a single technique [20]. Participants were ordered by recruitment date, and their group assignment was resampled at random 100,000 times using the design randomisation procedure and

assuming no treatment difference (S2 Appendix). The *p*-value was determined by the proportion of times that more extreme results than true results were observed by chance. Due to the minor change in randomisation ratio, the treatment effect and associated 95% confidence intervals were calculated using standard methods. For hypothesis testing, we report both standard *p*-values and a *p*-value that adjusts for the adaptive nature of the randomisation ratio as explained above.

For secondary efficacy analyses, all continuous variables were summarised using descriptive statistics depending on normality, specifically *n* (non-missing sample size), mean, SD, median, and IQR. Frequency and percentages were reported for categorical measures. Data were reported using confidence intervals and *p*-values where appropriate. The chi-squared test or Fisher exact test were used for categorical data and *t* tests or Mann–Whitney U tests for continuous data. The full trial protocol is appended to this manuscript (S1 Protocol). Trial enrolment, allocation, and follow-up were reported according to the adaptive designs CONSORT extension statement (S1 Checklist) [21].

## Results

### Interim results

A mean reduction in r-time of 1.86 minutes between the intervention and control arms was observed at interim analysis. The test statistic (−2.79) was below the predefined futility boundary ($u_1 = 1.8$). There was no significant difference in reoperation for bleeding ($p = 0.34$). Investigators were therefore allowed to recruit the second half of the study cohort. The Shapiro–Wilk test for normality was performed as stipulated by the study protocol; we found that the distribution of the primary outcome deviated significantly from normality (intervention, $p = 0.0061$; control, $p = 0.011$), thus triggering our adaptive decision rules based on an exponential distribution (S2 Appendix). As the relative difference between arms was 21.1% and therefore greater than 7.5%, we adapted the randomisation ratio for the remaining participants to be 1:1.33 as per the design.

### Full study results

**Participant characteristics.** Participants were recruited between April 2018 and August 2019 with interim analysis in May 2019. Of 520 patients screened, 228 met study inclusion criteria, consented to participate, and were randomised. Over the course of the study, 121 were allocated to the intervention group and 107 to the control group. A total of 7 participants were excluded from the intervention group due to inadvertent mechanical disruption during performance of the baseline or repeat TEG (Fig 1), and 1 participant in the intervention group was excluded due to lack of recorded data postoperatively. Participant characteristics, preoperative laboratory studies, and pre-CPB TEG results were similar when compared (Table 1).

**Heparin and protamine dosing.** Pre-CPB heparin dosing was similar in both groups (Table 2). A median reduction of 36.6% was observed in protamine dosing in the intervention group relative to the control group ($p < 0.001$), translating to a lower initial protamine:heparin ratio in the intervention group, based on the initial dose required to achieve a therapeutic ACT for CPB ($p < 0.001$). A similar difference was noted when the same ratio was recalculated for the total quantity of protamine administered relative to the initial dose of heparin when accounting for any additional protamine administered ($p < 0.001$). No difference was noted in the number of participants who required additional protamine within 4 hours postoperatively (10 [8.3%] versus 9 [8.4%]; $p = 1.00$).

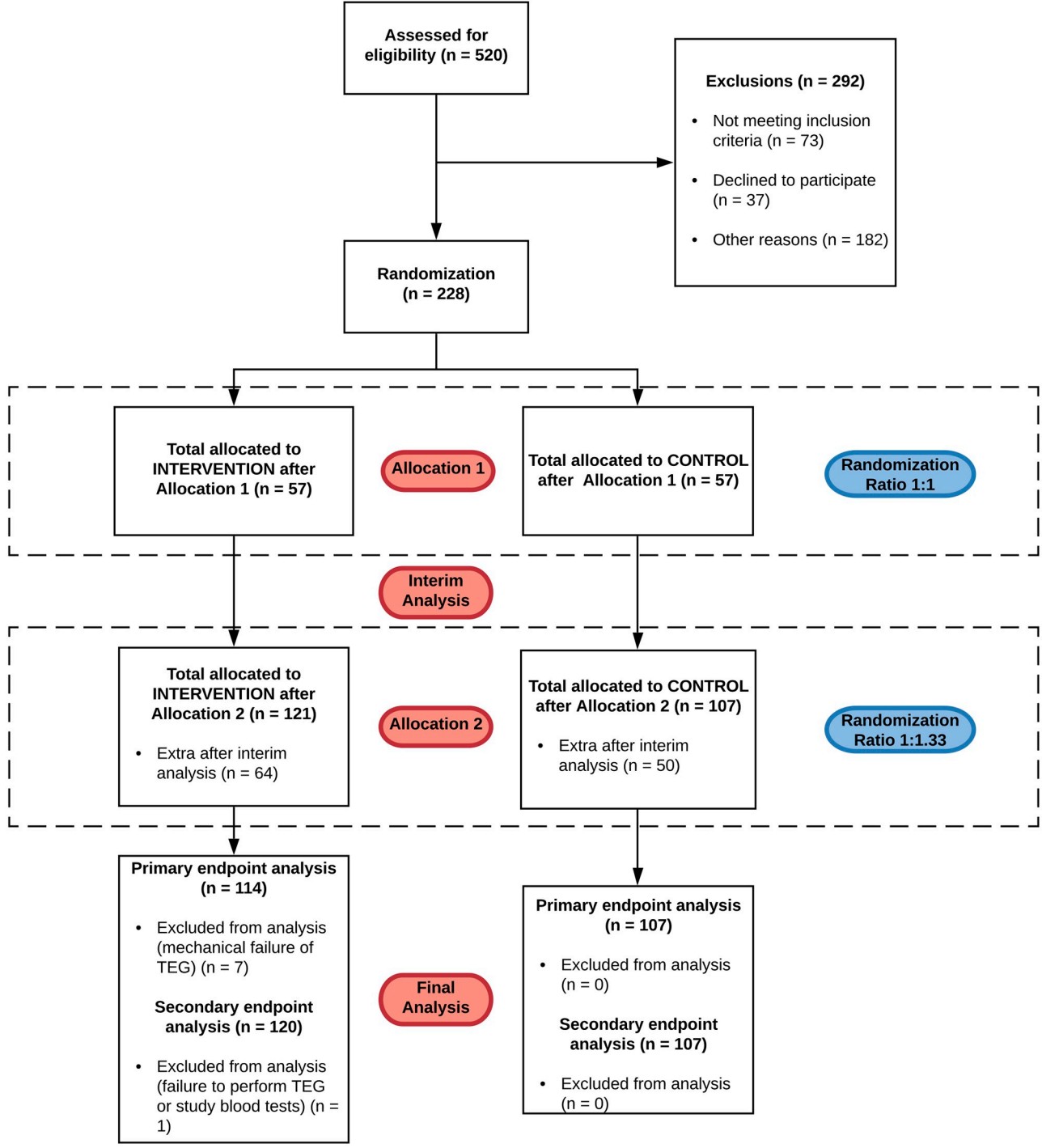

**Fig 1. ACE flow diagram of trial progress.** ACE, Adaptive designs CONSORT Extension; CONSORT, Consolidated Standards of Reporting Trials; TEG, thromboelastography.

**Primary endpoint.** The results from 221 participants were analysed for the primary endpoint (intervention group, $n = 114$; control group, $n = 107$). The mean TEG r-time measured 3 minutes post-protamine was significantly shorter in the intervention group, relative to the control group (6.58 [SD 2.5] versus 8.08 [SD 4.0] minutes; $p = 0.0016$; standard $p = 0.0011$).

**Table 1. Preoperative characteristics and covariates of participants allocated to the intervention (PRODOSE model) and control (fixed 1:1 ratio) groups.**

| | | Intervention (*n* = 121) | Control (*n* = 107) |
|---|---|---|---|
| Female sex (*n* [%]) | | 30 (24.8%) | 24 (22.4%) |
| Age, years | | 71.0 [61.0–77.0] | 70.0 [63.5–75.5] |
| BMI (kg m$^{-2}$) | | 28.2 [25.8–31.2] | 28.6 [26.0–31.7] |
| BSA (m$^2$) | | 1.92 (0.18) | 1.98 (0.19) |
| Weight (kg) | | 81.75 (13.6) | 85.6 (14.3) |
| EuroSCORE II | | 1.46 [0.85–2.18] | 1.33 [0.84–2.23] |
| Preoperative aspirin (*n* [%]) | | 97 (80.2%) | 85 (79.4%) |
| Hb concentration (g l$^{-1}$) | | 137.5 (13.1) | 137.1 (12.34) |
| Platelet count (× 10$^9$ l$^{-1}$) | | 237 (69) | 227 (63) |
| aPTT (s) | | 29.2 [28.1–31.4] | 29.2 [27.5–31.0] |
| PT (s) | | 12.0 [11.3–12.6] | 11.6 [11.0–12.1] |
| Creatinine (µmol/L) | | 80 [70–93] | 79 [67–93] |
| Pre-CPB kaolin r-time (min) | | 6.07 (2.13) | 5.69 (1.89) |
| Pre-CPB kaolin k-time (min) | | 1.39 (0.60) | 1.39 (0.41) |
| Pre-CPB kaolin α-angle (°) | | 70.81 (5.95) | 70.80 (5.82) |
| Pre-CPB kaolin MA (mm) | | 65.87 (5.62) | 64.50 (4.68) |
| Operation type (*n* [%]) | Ascending aorta | 0 | 3 (2.8) |
| | CABG | 67 (55.4) | 54 (50.5) |
| | CABG + ascending aorta | 1 (0.8) | 0 |
| | CABG + valve | 10 (8.3) | 18 (16.8) |
| | Valve | 37 (30.6) | 29 (27.1) |
| | Valve + ascending aorta | 6 (5.0) | 3 (2.8) |
| Lowest temperature on CPB (°C) | | 33.8 (1.46) | 33.8 (1.66) |
| CPB time (min) | | 86 [71–106] | 92 [71–118] |

Values are number (proportion), mean (SD), and median [IQR].

aPTT, activated partial thromboplastin time; BMI, body mass index; BSA, body surface area; CABG, coronary artery bypass grafting; CPB, cardiopulmonary bypass; Hb, haemoglobin; PT, prothrombin time.

The resultant standard treatment effect was a 1.5 (95% CI 0.61 to 2.39) minute reduction in post-protamine TEG r-time in the intervention group relative to the control group.

**Secondary endpoints.** While the r-time in the intervention group was more closely approximated to pre-CPB, a 25% increase in r-time post-CPB in the control group relative to

**Table 2. Unfractionated heparin and protamine dosing of participants allocated to the intervention (PRODOSE model) and control (fixed 1:1 ratio) groups.**

| | Intervention (*n* = 121) | Control (*n* = 107) | *p*-value |
|---|---|---|---|
| Initial heparin dose (×10$^3$ IU) | 27.0 [25.0–30.0] | 28.0 [25.0–30.0] | 0.098 |
| Additional heparin required during CPB (*n* [%]) | 56 (46.7) | 56 (52.3) | 0.47 |
| Total heparin dose (×10$^3$ IU) | 35.0 [30.0–40.0] | 36.0 [32.0–41.0] | 0.249 |
| Initial protamine dose (mg) | 180 [160–210] | 280 [250–300] | <0.001 |
| Initial heparin:protamine ratio | 0.66 [0.59–0.75] | 1.0 [1.0–1.0] | <0.01 |
| Total protamine dose (mg) | 210 [180–250] | 310 [283–360] | <0.001 |
| Total heparin:protamine ratio | 0.58 [0.53–0.65] | 0.86 [0.79–1.0] | <0.001 |

Values are number (proportion), mean (SD), and median [IQR].

CPB, cardiopulmonary bypass.

**Table 3. Secondary endpoints for participants allocated to the intervention (PRODOSE model) and control (fixed 1:1 ratio) groups.**

|  | Intervention ($n$ = 121) | Control ($n$ = 107) | $p$-value |
|---|---|---|---|
| Pre-CPB:post-CPB ACT ratio | 0.98 [0.91–1.07] | 0.99 [0.91–1.08] | 0.847 |
| Pre-CPB:post-CPB kaolin r-time ratio | 0.96 [0.78–1.14] | 0.75 [0.57–0.99] | <0.001 |
| Post-CPB heparinase r-time:post-CPB kaolin r-time ratio | 1.02 [0.95–1.12] | 1.02 [0.94–1.12] | 0.747 |
| Postoperative mediastinal/pleural drainage (mL) | 140 [75–245] | 135 [94–222] | 0.849 |
| Intraoperative PRBC requirement ($n$ [%]) | 11 (9.2%) | 9 (8.4%) | 1.0 |
| Number of PRBC transfused intraoperatively (units) | 2 [1–2] | 2 [1–2] | 0.967 |
| Postoperative PRBC requirement ($n$ [%]) | 19 (15.8%) | 14 (13.1%) | 0.691 |
| Number of PRBC transfused postoperatively (units) | 1 [1–2] | 1 [1–2.5] | 0.719 |

Values are number (proportion) and median [IQR]. Postoperative mediastinal/pleural drainage was measured at 4 hours. Postoperative RBC requirement was measured at 24 hours.

ACT, activated clotting time; CPB, cardiopulmonary bypass; PRBC, packed red blood cell.

baseline was noted ($p$ < 0.001) (Table 3). No difference in the ratio of post-CPB heparinase TEG to kaolin TEG ratio ($p$ = 0.747) or in the ratio of ACT pre-CPB to post-CPB ($p$ = 0.847) was seen in the intervention group, relative to the control group. We detected no evidence of a difference in mediastinal/pleural drainage at 4 hours postoperatively in the intervention group relative to the control group ($p$ = 0.85). Similarly, no evidence of difference was demonstrated in requirement for allogeneic red blood cell transfusion in the first 24 hours postoperatively ($p$ = 0.69) or the number of units of PRBCs administered intraoperatively ($p$ = 0.967) or postoperatively ($p$ = 0.719).

**Additional data.** Following administration of protamine, no clinically or statistically significant differences were demonstrated in the intervention group relative to the control group in mean haemoglobin concentration (100.7 [SD 13.7] g/L versus 101.7 [SD 11.7] g/L; $p$ = 0.559), median platelet count (145 [IQR 116 to 178] $\times 10^9$/L versus 134 [112 to 166] $\times 10^9$/L; $p$ = 0.153), median aPTT (39.5 [IQR 34.2 to 44.8] seconds versus 37.9 [IQR 32.8 to 42.1] seconds $p$ = 0.123), median PT (15.9 [IQR 14.7 to 16.8] seconds versus 16.0 [IQR 15.0 to 16.9] seconds; $p$ = 0.529), or median fibrinogen concentration (2.0 [IQR 1.7 to 2.4] g/L versus 2.1 [IQR 1.7 to 2.3] g/L; $p$ = 0.559). No differences were noted in requirement for fresh frozen plasma or prothrombin complex concentrate (19 [15.8%] versus 14 [13.1%]; $p$ = 0.221), platelets (5 [4.2%] versus 2 [1.9%]; $p$ = 0.451), or cryoprecipitate/fibrinogen concentrate (0 [0%] versus 3 [2.8%]; $p$ = 0.103) up to 24 hours postoperatively.

Post-CPB TEG data are shown in Table 4 (in addition to the stated primary and secondary outcomes). These were performed as post hoc exploratory analyses. The intervention group was statistically superior in a variety of metrics, including kaolin k-time ($p$ = 0.013), α-angle ($p$ = 0.002) and maximum amplitude (MA) ($p$ = 0.05), and heparinase r-time ($p$ = 0.003), k-time ($p$ = 0.01), and α-angle ($p$ = 0.006).

**Safety endpoint.** No difference was noted in rate of reoperation for bleeding in the first 24 hours after the primary operation (3 [2.5%] versus 3 [2.8%]; $p$ = 1.00).

## Discussion

We conducted an adaptive, open-label, randomised superiority trial of mathematical model-based protamine dosing versus a standardised 1:1 fixed ratio following CPB in patients presenting for non-emergent cardiac surgery. We found that model-based dosing led to superior in vitro clot kinetics in the post-CPB period as measured by kaolin TEG r-time and that

**Table 4. Kaolin and heparinase post-CPB TEG results for participants allocated to the intervention (PRODOSE model) and control (fixed 1:1 ratio) groups.**

| | Intervention ($n$ = 121) | Control ($n$ = 107) | $p$-value |
|---|---|---|---|
| Post-CPB kaolin k-time (min) | 1.56 (0.52) | 1.93 (1.49) | 0.013 |
| Post-CPB kaolin ɑ-angle (°) | 69.23 (5.66) | 66.14 (8.45) | 0.002 |
| Post-CPB kaolin MA (mm) | 61.70 (6.16) | 60.07 (6.12) | 0.05 |
| Post-CPB heparinase r-time (min) | 6.63 (3.00) | 7.99 (3.72) | 0.003 |
| Post-CPB heparinase k-time (min) | 1.53 (0.62) | 1.80 (0.94) | 0.01 |
| Post-CPB heparinase ɑ-angle (°) | 69.08 (7.40) | 66.03 (8.92) | 0.006 |
| Post-CPB heparinase MA (mm) | 59.60 (6.08) | 58.75 (6.75) | 0.325 |

Values are mean (SD).

CPB, cardiopulmonary bypass; MA, maximum amplitude; TEG, thromboelastography.

participants in the intervention group had a viscoelastic profile that more closely mirrored their pre-CPB condition. Furthermore, we found that this result was achieved with a median 36.6% reduction in protamine dose. Despite this reduction in protamine dose, we found no evidence of clinically or statistically significant difference in postoperative bleeding or transfusion requirement.

Debate regarding the best numerical ratio has been ongoing for some time [22,23] and has spurred the development of other techniques for optimised protamine dosing such as point-of-care, individualised titration devices [24,25]. However, these are expensive, and their performance is mixed [23–27]. Pharmacokinetic algorithms have been promoted as a cheaper, more readily accessible alternative. Previous retrospective observational [13,28], prospective observational [12,29–31], and pilot randomised controlled trials [32,33] have examined different mathematical models for determining heparin concentration following CPB. Kjellberg and colleagues have performed the sole, adequately powered randomised trial in the literature that compares such a model to a fixed ratio approach [10]. Like our findings, this study demonstrated a reduction in the amount of protamine required at the conclusion of CPB but did not show a statistically significant difference in blood loss or transfusion.

It must be acknowledged that this trial was not designed or powered to detect a difference in clinical endpoints, and type II error cannot be excluded. Our exclusion criteria were chosen to minimise the risk of postoperative bleeding so that the effect of the intervention on TEG r-time could be investigated without the potential confounding effects of major haemorrhage that is inherent to certain cardiac surgical procedures and patients. A similar trial in a cardiac surgical population at higher risk for bleeding (i.e., complex aortic surgery and/or preoperative heparin infusion), powered specifically to look at these outcomes, should be considered.

There are a number of distinctions between our study and this previous work [10]. Notably, Kjellberg and colleagues did not quantitatively measure any coagulation function (other than ACT) as an endpoint, limiting their analysis to bleeding and transfusion. We chose viscoelastic testing as our primary outcome measure for 3 reasons: Firstly, TEG and similar devices are able to detect in vitro abnormalities at far lower protamine concentrations than ACT [17], and, in some prospective studies, are more predictive of postoperative bleeding than conventional coagulation tests [34]. We acknowledge that the relationship between deranged viscoelastic metrics and bleeding is inconsistent when multivariate analysis is performed [35]. Secondly, following the publication of Karkouti and colleagues [36], viscoelastic testing now forms the basis of many integrated transfusion algorithms in cardiac surgery, and, therefore,

the incorporation of TEG into our study, combined with easily measured and clinically relevant secondary outcomes, make our results more generalisable (within the limits of the study design and exclusion criteria). Finally, TEG r-time was the primary outcome in the studies whose findings we most closely sought to confirm [4,13]. Another point of differentiation with Kjellberg and colleagues is our use of an adaptive design. This adaptive randomisation is a significant advantage—as well as a check for futility and safety; the interim analysis allowed adaptation of the randomisation ratio to recruit a higher number of patients to the arm trending towards benefit while preserving study power.

We acknowledge some limitations. Firstly, the surgeon and anaesthetist were not blinded to the group allocation. Any potential bias related to this was minimised by standardisation of coagulation management in the pre-CPB period, in the immediate post-CPB period, and in the first 24 hours postoperatively, and by the selection of quantitative rather than qualitative endpoints. Secondly, our model does not correct for the effects of possible hypothermia during CPB. Hypothermic CPB decreases heparin requirements [37], likely as a consequence of reduced heparin metabolism at lower temperatures [38]. This problem will not be resolved until the rate of heparin metabolism at differing temperatures is better defined and will require a continuous temperature input into any associated model. Thirdly, we did not measure intraoperative blood loss, as this is challenging to do accurately during cardiac surgery [39]. Kjellberg and colleagues did attempt to measure intraoperative blood loss and detected no statistically significant median difference (300 [IQR 200 to 500] versus 400 [IQR 250 to 500] mL; $p$ = 0.219). Fourthly, our study made use of 2 types of TEG device depending on the recruiting centre (TEG 5000 or TEG 6s). The concordance between these devices appears consistent in the cardiac surgical setting [40]. Additionally, the study exclusion criteria mean the results cannot be generalised to certain groups (notably, obese patients, children, and those with end-stage kidney disease requiring dialysis). These patients should be the subject of further prospective validation. Finally, we did not make a further, formal assessment of coagulation beyond the 3-minute mark following protamine administration. Protamine is recognised as having a short biological half-life, with elimination from the circulation within 5 minutes [41]. Ergo, we cannot be certain that the observed effect on TEG r-time would have persisted beyond this initial measurement. However, the anticoagulant and antiplatelet effects of protamine are known to persist beyond this time [42], and increased protamine-to-heparin ratio has been linearly associated with prolonged postoperative bleeding [9].

We conclude that the PRODOSE model delivers improved in vitro metrics of clot firmness (specifically, our primary endpoint, TEG r-time) when compared to a fixed 1:1 ratio based on the initial dose of heparin required to achieve a therapeutic ACT for CPB. The model reduces the dose of protamine required by a median of 36.6%. In this population, this approach appeared safe. Further studies are necessary to determine whether there is a clinical benefit to this approach in patients undergoing cardiac surgery at higher risk for bleeding.

## Supporting information

**S1 Appendix. Trial guidance for operating theatre and ICU management of bleeding and coagulopathy.** ICU, intensive care unit.
(DOCX)

**S2 Appendix. Supporting information methods.**
(DOCX)

**S1 Protocol. Trial protocol and protocol amendments.**
(PDF)

**S1 Checklist. ACE statement.** ACE, Adaptive designs CONSORT Extension; CONSORT, Consolidated Standards of Reporting Trials.
(DOCX)

# Acknowledgments

The authors thank the Jon Moulton Charity Trust and UK Medical Research Council for the financial support necessary to conduct this study. The trial was conducted with the assistance of the Papworth Trials Unit Collaboration; special thanks are due to Mr Thomas Devine and Ms Carol Freeman for their support in project and data management. The authors also acknowledge the data management support provided by Ms Sofia Sidiropoulos, RN; Ms Sarah Bauch, RN; Ms Gayle Claxton, RN; and Ms Saskia Harris, RN of the Department of Anaesthesia, Austin Health.

# Author Contributions

**Conceptualization:** Lachlan F. Miles, Ravi De Silva, Florian Falter.

**Data curation:** Lachlan F. Miles, Christiana Burt, Joseph Arrowsmith, Mikel A. McKie, Sofia S. Villar, Pooveshnie Govender, Ruth Shaylor, Zihui Tan.

**Formal analysis:** Mikel A. McKie, Sofia S. Villar.

**Funding acquisition:** Florian Falter.

**Investigation:** Lachlan F. Miles, Christiana Burt, Joseph Arrowsmith, Florian Falter.

**Methodology:** Lachlan F. Miles, Mikel A. McKie, Sofia S. Villar, Florian Falter.

**Project administration:** Florian Falter.

**Resources:** Ruth Shaylor, Zihui Tan, Florian Falter.

**Supervision:** Lachlan F. Miles, Christiana Burt, Joseph Arrowsmith, Mikel A. McKie, Sofia S. Villar, Ravi De Silva, Florian Falter.

**Validation:** Lachlan F. Miles, Joseph Arrowsmith, Mikel A. McKie, Sofia S. Villar, Pooveshnie Govender, Florian Falter.

**Visualization:** Florian Falter.

**Writing – original draft:** Lachlan F. Miles, Christiana Burt, Mikel A. McKie, Sofia S. Villar, Florian Falter.

**Writing – review & editing:** Lachlan F. Miles, Christiana Burt, Joseph Arrowsmith, Mikel A. McKie, Sofia S. Villar, Pooveshnie Govender, Ruth Shaylor, Zihui Tan, Ravi De Silva, Florian Falter.

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
