## [Editor Report · Decision Letter 0]

11 Nov 2020

Dear Dr Miles, 

Thank you for submitting your manuscript entitled "Optimal protamine dosing after cardiopulmonary bypass: The PRODOSE adaptive randomised controlled trial" for consideration by PLOS Medicine.

Your manuscript has now been evaluated by the PLOS Medicine editorial staff [as well as by an academic editor with relevant expertise] and I am writing to let you know that we would like to send your submission out for external peer review.

Kind regards,

Adya Misra, PhD,

Senior Editor

PLOS Medicine

---

## [Decision Letter · Decision Letter 1]

7 Dec 2020

Dear Dr. Miles,

Thank you very much for submitting your manuscript "Optimal protamine dosing after cardiopulmonary bypass: The PRODOSE adaptive randomised controlled trial" (PMEDICINE-D-20-05286R1) for consideration at PLOS Medicine. 

Your paper was evaluated by a senior editor and discussed among all the editors here. It was also discussed with an academic editor with relevant expertise, and sent to independent reviewers, including a statistical reviewer (r#2). The reviews are appended at the bottom of this email and any accompanying reviewer attachments can be seen via the link below:

[LINK]

In light of these reviews, I am afraid that we will not be able to accept the manuscript for publication in the journal in its current form, but we would like to consider a revised version that addresses the reviewers' and editors' comments. Obviously we cannot make any decision about publication until we have seen the revised manuscript and your response, and we plan to seek re-review by one or more of the reviewers. 

We expect to receive your revised manuscript by Dec 28 2020 11:59PM. Please email us (plosmedicine@plos.org) if you have any questions or concerns.

We look forward to receiving your revised manuscript. 

Sincerely,

Emma Veitch, PhD

PLOS Medicine

On behalf of Adya Misra, PhD, Senior Editor, 

PLOS Medicine

plosmedicine.org

*Please structure the abstract using the PLOS Medicine headings (Background, Methods and Findings, Conclusions - "Methods and findings" is a single subsection). In the last sentence of the Abstract Methods and Findings section, please include a brief note about any key limitation(s) of the study's methodology.

*As recommended by one reviewer, we'd recommend you update the abstract to explain a bit better how the adaptive nature of the trial design modified the design at the interim analysis (this does not need to go into substantial detail). We'd also ask that all the statistical presentations clearly indicate what the different figures show (this is mainly a problem for the numbers given in square brackets, which we assumed were 95% CI but should be indicated).

*Please reformat the citation style into PLOS Medicine's format (should be straight forward if using referencing software) - this should use callouts formatted as sequential numerals in square brackets (not superscript or round brackets).

*At this stage, we ask that you include a short, non-technical Author Summary of your research to make findings accessible to a wide audience that includes both scientists and non-scientists. The Author Summary should immediately follow the Abstract in your revised manuscript. This text is subject to editorial change and should be distinct from the scientific abstract. Please see our author guidelines for more information: https://journals.plos.org/plosmedicine/s/revising-your-manuscript#loc-author-summary

*We'd ask the authors to clarify whether the secondary outcomes as set out in the manuscript match up against what is set out in the trial registry record (https://clinicaltrials.gov/ct2/show/NCT03532594) which specifies that the secondary outcomes include blood loss at four hours post-surgery as measured with intercostal drain output, and blood products usage at 24 hours. If not, please make sure all analyses for prespecified secondary outcomes (as set out in the original protocol) are included in the published paper. If the trial registry record needs to be updated, that can be done retrospectively. 

*Many thanks for using the ACE guideline to support trial reporting - please also append (as a supporting information file) the ACE checklist, which is available to download from https://trialsjournal.biomedcentral.com/articles/10.1186/s13063-020-04334-x#Sec5 (please complete with section/para's)

*Per the author guidelines for papers reporting results of a trial (https://journals.plos.org/plosmedicine/s/submission-guidelines#loc-clinical-trials), please also upload as supporting information a copy of the original trial protocol for the PRODOSE trial. 

Comments from the reviewers:

Reviewer #1: 

General comments: The authors are congratulated for validating the previously suggested concept that the conventional wisdom of giving protamine at a 1:1 ratio to heparin might induce a transient protamine overdose. The authors used TEG R-time as a primary endpoint, and reported that the intervention group which had a median ratio of 0.66 had a shorter R-time post-protamine when compared to the control which had a median ratio of 1.0. However, the authors failed to demonstrate that the intervention lowers post-operative bleeding or allogeneic blood transfusion. The 'excess' dose may not have had a prolonged impact on clinical hemostasis because protamine disappears rapidly (<5 min) from circulation (Butterworth J, et al. Ann Thorac Surg 2002). The authors obtained the TEG samples in 3 min after protamine administration, which demonstrated a transient negative impact on the TEG R-time. 

Overall, the manuscript was well written, but additional clarifications are needed. 

Specific comments:

P5, Patients were excluded if they were < 18 years old, had a total body weight > 120 kg (due to unpredictable heparin requirements in obese individuals) (12), were dialysis dependent

--The reviewer feels that protamine dosing mistakes can have more serious consequences in the excluded patients. This was mentioned briefly that their findings cannot be "generalized", but they should rephrase that these patients should be more closely studied. 

P5, unfractionated heparin infusion 

--The reviewer does not fully understand why the authors excluded the patients on heparin infusion before surgery. Again, the impact of protamine management is more likely to be important in complex cases. 

P9, Using a control post-protamine r-time of 4.2 minutes, a standard deviation of 1.27 minutes and a minimum clinically relevant effect size of 15%

--The definition of clinical relevance is unclear. Why is a 1.27-min difference clinically important? On page 30, the authors stated that FFP or PCC is given in the case of R-time above 8 min, which is far from 4.2 +/- 1.27 min. 

P11, As the relative difference between arms was greater than 7.5% (21.1%), we adapted the randomisation ratio for the remaining participants to be 1:1.33 as per the design

--Were the authors aware that the intervention groups had a greater than 7.5%, suggesting a superiority at the preliminary evaluation point? It was unclear how the authors and statisticians decided to make a 1:1.33 ratio. Please revise the sentences, so that readers can clearly understand the sample size changes, and their justifications. 

P14, Despite this reduction in protamine dose, we found no evidence of clinically or statistically significant difference in post-operative bleeding or transfusion requirement.

--The authors chose relatively straight forward cases, and thus the overall impact on clinical endpoints might have been minimal. Secondly, the control group received a 100-mg more protamine than in the intervention group. This 'excess' dose may not have had a prolonged impact on clinical hemostasis because protamine disappears rapidly (<5 min) from circulation (Butterworth J, et al. Ann Thorac Surg 2002). The authors obtained the TEG samples in 3 min after protamine administration, which demonstrated a transient negative impact on the TEG R-time. This possibility should be mentioned in the discussion. 

P27, Table 3

--What were the duration of observation for 'Postoperative mediastinal drainage' and 'Postoperative RBC requirement'? 

P32, At this point the effect size was assessed having noted a greater than 7.5% difference in treatment arms, the second 114 group assignments was sampled at a ratio of 1:1.33. If a difference of this magnitude had not been noted, the ratio would have remained at 1:1 

--Please explain why a greater than 7.5% was used as a cut-off to increase the sample size of the intervention when it already appeared to be superior? What would have happened if this was not changed? 

Reviewer #2: 

This is a statistical review of manuscript PMEDICINE-D-20-05286R1. There are a couple of points that require further clarification.

Abstract

1/ I think that a bit more explanation on what is meant by "adaptive" would be good. Perhaps "two-stage with revised randomisation ratio" or something along these lines. 

2/ What do you report in the "[]". Are these 95% CIs, or SDs, or SEs ? This comment is also valid for the main text.

Main text

1/ Trial design: "An interim analysis was scheduled to check for safety, consider a pre-defined futility rule and to adapt the randomisation ratio based on the primary outcome data at that point". I agree that all details are in Appendix 2. However, some information needs to come in the main text, including the potentially revised randomisation ratio, perhaps the thresholds for futility etc 

2/ Statistical analysis; "Primary efficacy analysis was carried with a re-randomisation-based method, ensuring type I error was preserved despite deviations in assumptions and taking the adaptive design into account." Please could you clarify why you used this re-randomisation method? Was it not possible to use the Mann-Whitney test (or something equivalent), perhaps even after transformation of the primary endpoint if not normally distributed? 

3/ Results: 

"We found that the distribution of the primary outcome deviated significantly from normality (intervention, p = 0.0061; control, p = 0.011)". What are these p-values ? Test for normality ? If yes which test ?

"As the relative difference between arms was greater than 7.5% (21.1%)" So what is 21.1%, the observed relative difference at that point ? 

Reviewer #3: 

This interesting manuscript addresses the clinically challenging dosing scheme of protamine, in order to reverse the heparin effect after CPB in cardiac surgical patients.

The main question remains whether any of the interventions and results in this manuscript are novel.

Specific Comments

Abstract

1) Methods, please include how many patients were in the treatment/model group and how many in the fixed ratio group.

2) Results, the authors should clarify the type of transfusion and units.

3) Conclusion, a comment/conclusion should be added that there was no difference between groups in blood loss and blood transfusion.

Introduction

4) The recent EACTS/EACTA/EBCP guidelines (Wahba et al. 2020) recommend individualised heparin and protamine management in order to reduce postoperative coagulation abnormalities and bleeding complications in cardiac surgery with CPB. This should be added to the second paragraph, particularly as it deviates from a fixed (1:1) dosing recommendation from earlier guidelines (Pagano et al. 2016).

5) The Kjellberg study, introducing an algorithm to calculate protamine in a randomised controlled trial, should be listed in the second paragraph of the introduction (reference 31). The results of this RCT revealed that there was a reduced dose of protamine, but blood loss and transfusion rates were similar in the intervention group when compared to the control group.

Methods

6) Please clarify the type of heparin used and the manufacturer. Different manufacturers may produce heparins with different coagulation effects.

Results

7) What technique of residual blood management was used at the end of CPB: direct retransfusion of unprocessed blood, retransfusion of processed blood or both? This may have an effect on coagulation factors administered to patients at the end of CPB and should therefore be clarified, particularly if there were difference between groups.

8) Please include actual perioperative results of aPTT, PT, fibrinogen, ATIII levels, kaolin and heparinase TEGs and platelet counts in the results section, in addition to the presented p-values.

9) The number of FFPs and platelet concentrates administered in patients of each group should be added. 

10) Was there any difference between PRC transfusions between groups beyond 24hrs? 

Discussion

11) In order to address reversal of heparin it would also be important to measure serum concentrations of heparin levels after the protamine administration. This limitation should be discussed.

12) How does the protamine dose algorithm compare with individualised point-of-care devices (e.g. Hepcon) for protamine dosing? What are potential advantages of the Hepcon method of titrating heparin and protamine? Should Hepcon guided coagulation management in cardiac surgery be compared with a protamine dose algorithm in the future?

13) It should be discussed if there is a potential clinical benefit for patients if protamine dose algorithms are used?

[LINK]

---

## [Decision Letter · Decision Letter 2]

7 May 2021

Dear Dr. Miles,

Thank you very much for re-submitting your manuscript "Optimal protamine dosing after cardiopulmonary bypass: The PRODOSE adaptive randomised controlled trial" (PMEDICINE-D-20-05286R2) for consideration at PLOS Medicine. We do apologize for the long delay in sending you a decision. 

I have discussed the paper with editorial colleagues and our academic editor, and it was also seen again by one reviewer. I am pleased to tell you that, provided the remaining editorial and production issues are fully dealt with, we expect to be able to accept the paper for publication in the journal.

[LINK]

Please let me know if you have any questions in the meantime, and we look forward to receiving the revised manuscript shortly.   

Sincerely,

Richard Turner, PhD

rturner@plos.org

Requests from Editors:

Please adapt the wording of the data statement to "... cannot be made publicly available ..." or similar (assuming this is the case). 

It seems that the trial registration was finalized on May 22, 2018. Please explain the short delay after the quoted start date for recruitment. In view of the dates, you may wish to remove the word "prospectively" (registered) in the first paragraph of your Methods section. 

Please quote the study dates in your abstract.

Please specify the study's primary endpoint in your abstract, we suggest around line 15. We suggest also adding a subsequent "Secondary endpoints included ..." sentence, quoting the secondary endpoints on which you report findings in the abstract.

Please adapt the text around line 22 of the abstract to indicate that this is the primary study finding. 

Regarding the abstract and main text, we are used to seeing an effect size with associated 95% CI quoted for the primary endpoint of a trial. Is this something you can provide?

We suggest quoting the findings on the safety endpoint immediately prior to the final "limitations" sentence of the "Methods and findings" subsection of your abstract. 

Please quote the trial registration number in your abstract.

Please adapt the format of your author summary so that the 3 subsections each consist of about 3 bulleted points (each of 1-2 short sentences). You may find it helpful to consult one or two recent research papers in PLOS Medicine to get a sense of the preferred style. 

Please use the active voice (e.g., "We tested ...") in one or two points in the author summary.

Please rename the attached CONSORT checklist "S4_CONSORT_Checklist" or similar, and refer to it by this label in the Methods section (we suggest using the acronym "CONSORT" in the text, incidentally). 

Please adapt the checklist so that individual items are referred to by section (e.g., "Methods") and paragraph number. Please do not use page or line numbers, as these will change in the published paper. 

The first paragraph of the discussion should summarize the study. Therefore, you may wish to truncate this paragraph immediately prior to "However, it must be acknowledged ..." and the subsequent sentences could be moved to other paragraphs in which related literature and limitations are discussed. 

Please revisit the discussion of blinding throughout the text, to ensure consistency. It seems that the participants were blinded, and so it may help to add a few words to explain the statement "effectively unblinded" (Discussion section).

Please remove the information on data availability, funding and competing interests from the end of the main text. This will appear in the article metadata, via entries in the submission form. 

Please remove "table of contents" and the PubMed details from reference 41. 

Comments from Reviewers:

*** Reviewer #2: 

I thank the authors for satisfactorily addressing my previous comments, and don't have further comments.

***

[LINK]

---

## [Editor Report · Decision Letter 3]

14 May 2021

Dear Dr Miles, 

On behalf of my colleagues and the Academic Editor, Dr Rahimi, I am pleased to inform you that we have agreed to publish your manuscript "Optimal protamine dosing after cardiopulmonary bypass: The PRODOSE adaptive randomised controlled trial" (PMEDICINE-D-20-05286R3) in PLOS Medicine.

Prior to acceptance please address two small issues:

- please check that all numbers are quoted consistently throughout the paper (we think that "0.96 [IQR 0.78 - 1.04]" is quoted as "0.96 [IQR 0.78 - 1.14]" in the table 3);

- the numbering of the study protocol and CONSORT attachments are reversed at the end of the Methods section.

PRESS

Sincerely, 

Richard Turner, PhD 

rturner@plos.org